# [Rp] Learning to Deceive with Attention-Based Explanations

**Jelle van den Broek**
jelle.vandenbroek@student.uva.nl
Student ID: 11882425

**Koen Gommers**
koen.gommers@student.uva.nl
Student ID: 11871067

**Jeroen Taal**
jeroen.taal@student.uva.nl
Student ID: 11755075

**Dennis Swart**
dennis.swart@student.uva.nl
Student ID: 11892153

## Reproducibility Summary

**Scope of Reproducibility**

Learning to Deceive with Attention-Based Explanations by Pruthi et al. makes two claims which we reproduce in this work. Their first claim entails that attention weights can be manipulated to shift mass away from a predefined set of impermissible tokens without significant chance in performance. Their second claim entails that these manipulated models still rely on information from the set of impermissible tokens. In this reproducibility report we argue that by running their experiments, we can reproduce the results which support the claims by Pruthi et al..

**Methodology**

In this report we run three of the classification tasks used by Pruthi et al., as the Reference Letters dataset is not publicly available. We extended this part by adding the multiclass version of the sentiment classification dataset. We evaluated these tasks with a Embedding model a BiLSTM and a BERT. In addition, we also run the experiments using the four sequence-to-sequence tasks with a GRU.

**Results**

Most of the results were reproducible with a difference of less than 1% in accuracy of the original paper. Therefore, we argue that the claims made by Pruthi et al. could be supported by this report.

**What was easy**

The authors open-sourced their code and responded quickly to questions by e-mail. Furthermore, the experiments can be run in a reasonable amount of time on a GPU.

**What was difficult**

Although the code was open-sourced, the code for some experiments was missing, and for certain experiments instructions for running the code were missing. Furthermore, the dataset for one of the classification tasks, namely the Reference Letters dataset, is not publicly available.

**Communication with original authors**

We had brief contact with the original authors asking for missing code. They responded quickly and added missing code which they provided.

# 1 Introduction

Attention mechanisms, originally introduced by Bahdanau et al. (2015) to perform machine translation, are widely used in deep learning, such as in natural language processing, speech processing and computer vision. The attention weights in attention mechanisms are often used to show which input tokens the model attends to. As a result, these weights are also often used by practitioners to illustrate to stakeholders which input tokens affected the decision of the model. However, Pruthi et al. (2019) show that the attention weights can easily be manipulated during training without a significant change in performance.

They propose a method which manipulates the attention weights of a model during training. The attention weights are manipulated by introducing an additional penalty term which penalises a model for assigning attention to a pre-defined set of impermissible tokens. By comparing the performance and the attention mass assigned to the impermissible tokens of different models with their manipulated counterparts on both classification and sequence-to-sequence datasets, Pruthi et al. make the following claims:

- Attention weights can be manipulated to shift mass away from impermissible tokens without significant change in performance.

- Manipulated models still rely on information from impermissible tokens.

In this reproducibility report we run four classification tasks and four sequence-to-sequence tasks to test the claims made by Pruthi et al.. For the classification experiments, three attention-based models are trained and evaluated on four classification tasks. The four classification experiments consist out of three binary classification task and one multiclass classification task. For the sequence-to-sequence experiments, an encoder-decoder model with varying attention mechanisms is trained and evaluated on four tasks. Three of these tasks are toy datasets created by Pruthi et al., the fourth task is an English to German machine translation task (More information in section 3). Further on in the report, we display the results obtained by conducting these experiments and compare them to the results reported by Pruthi et al. (Section 4 & 5).

We cannot reproduce one of the binary classification tasks from the paper of Pruthi et al., because they do not have permission to share this private dataset. Therefore, we substitute this dataset for a multiclass classification dataset. As Wiegreffe and Pinter (2019) state, complex networks can produce outputs which can easily be aggregated to form the same binary prediction. Therefore we evaluate the models on a multiclass classification task harder task, to see whether the first claim also generalises to a more difficult multiclass task.

In addition to these classification and sequence-to-sequence tasks, Pruthi et al. also conduct a human study. In this study, three human subjects were asked to look at the attention mass of three different models and evaluate if these models made their decision based on impermissible tokens. Thereafter they were asked to score the trustworthiness of the model by evaluating if the attention mass of the model actually drove the decision of the model. We did not deem it necessary to include this experiment in this report, because none of the claims mentioned above are supported by this experiment.

# 2 Related Work

There has been more work put into investigating whether attention can be used to explain the predictions of a model. For example Jain and Wallace (2019), show that adversarial attention weights can be found after training without changing the predictions of the model. Furthermore, Jain and Wallace found that attention weights correlate weakly with other measures of feature importance. In other work, Serrano and Smith (2019) show that a significant portion of the highest attention weights can be set to zero without change predictions of a model.

In concurrent work to that of Pruthi et al., Wiegreffe and Pinter (2019) challenge the conclusions of Jain and Wallace (2019). They argue that attention might provide *an* explanation and not *the* explanation. Furthermore, they argue that Jain and Wallace did not find adversarial attention weights which are equally plausible as the learned attention weights. By changing the attention weights after training, these weights are treated as independent of the rest of the model, even though they are learned in combination with the other layers in the network.

Therefore Wiegreffe and Pinter introduce a training scheme which produces attention weights as dissimilar as possible to that of a base model. These weights are similar to those found by Jain and Wallace, however, now they are actually learned by the model during training. However, Wiegreffe and Pinter find that the learned adversarial attention weights still perform poorly relative to the attention weights of the base model.

However, Pruthi et al. claim to have proposed a training scheme which results in more deceptive adversarial attention weights. Where Wiegreffe and Pinter conduct further tests to investigate when attention could be used as an explanation, Pruthi et al. focus on showing that they are able to learn alternative attention weights with minimal cost in performance.

## 3 Methodology

In this section we will first present the attention manipulation scheme as introduced by Pruthi et al.. Thereafter we will briefly discuss the models on which this manipulation scheme was applied, and the datasets these models are evaluated on. At last, we will discuss any implementation details necessary for reproducing the results.

### 3.1 Attention manipulation

Pruthi et al. introduce a penalty term $\mathcal{R}$, which can be added to any task-specific loss function $\mathcal{L}$ to form a new objective function: $\mathcal{L}' = \mathcal{L} + \mathcal{R}$. Given a pre-defined of set impermissible tokens $\mathcal{I}$, and an input sentence $S = w_1, ..., w_n$, they define $\boldsymbol{\alpha} \in [0,1]^n$ as being the vector of attention weights assigned to the words in $S$. Furthermore, we can define an input mask $\mathbf{m}$:

$$\mathbf{m}_i = \begin{cases} 1, & \text{if } w_i \in \mathcal{I} \\ 0, & \text{otherwise} \end{cases}$$

as a binary vector indicating whether word $w_i$ is in the set of impermissible tokens. For a single attention layer, $\mathcal{R}$ can then be defined as:

$$\mathcal{R} = -\lambda \log(1 - \boldsymbol{\alpha}^T \mathbf{m}) \tag{1}$$

Where $\lambda$ is explained to represent a modulation penalty coefficient of attention assigned to impermissible tokens, and $(1 - \boldsymbol{\alpha}^T \mathbf{m})$ the total attention weights assigned to permissible tokens.

However, some models, such as the BERT model, which we describe in more detail in section 3.2.1, employ multiple attention mechanisms at each layer. Let $\mathcal{H}$ be the set of different attention mechanisms, often referred to as attention heads, at a layer. Pruthi et al. introduce two adjustments to the penalty term in 1. One which penalises the mean of attention assigned to impermissible tokens by the different heads (equation 2). The other penalises the maximum attention assigned to impermissible tokens (equation 3).

$$\mathcal{R} = -\frac{\lambda}{|\mathcal{H}|} \log(1 - \boldsymbol{\alpha}^T \mathbf{m}) \tag{2}$$

$$\mathcal{R} = -\lambda \min_{h \in \mathcal{H}} \log(1 - \boldsymbol{\alpha}^T \mathbf{m}) \tag{3}$$

### 3.2 Model descriptions

In this subsection we will describe the models on which the attention manipulation scheme was applied. Pruthi et al. employ two different sets of models, models used for classification tasks, and models used for sequence-to-sequence tasks. For the classification tasks they evaluated three models of differing complexity (number of parameters). For the sequence-to-sequence tasks Pruthi et al. employ a single encoder-decoder model, but evaluate this model using different attention mechanisms.

#### 3.2.1 Classification models

**Embedding + Attention**  Is the least complex model as it only consists of learned word-embeddings which are aggregated via a single dot product attention mechanism to form a single context vector. This context vector is then fed into a linear classifier to produce a distribution over classes in the dataset. This model is implemented using an embedding size of 128.

**LSTM + Attention**  Is a single layer bi-directional LSTM as introduced by Graves and Schmidhuber (2005). The input of the model is first embedded by learned embedding vectors of size 128. Thereafter, the embeddings are fed into a bi-directional LSTM which produces a context vector for each input token of size 128 * 2. These context vectors are aggregated via a dot product attention mechanism of which the query vectors are learned, resulting in a single context vector. The final context vector is fed into a linear layer with Softmax to produce a distribution over the classes in the dataset.

**BERT**   As the most complex model, the base version of Bidirectional Encoder Representations from transformers (BERT), introduced by Devlin et al. (2019), is used. This version consists of 12 layers each of which has 12 attention heads. As described in section 3.1, Pruthi et al. employ two different manipulation strategies for the BERT model which will be referred to as BERT (mean) (equation 2) and BERT (max) (equation 3).

Furthermore, as described by Devlin et al. (2019), each sequence is prepended with a [CLS] token, of which the hidden vector in the last layer is used as the sequence embedding. This hidden vector is fed into a linear classifier to perform classification. Pruthi et al. use the pretrained BERT$_{\text{BASE}}$ trained on uncased text data. The pretrained model and classifier are fine-tuned on the datasets described in section 3.3.

However, as this version of BERT contains 12 layers with attention mechanisms, it would be possible for information to flow between permissible and impermissible tokens. To prevent this, Pruthi et al. apply an attention mask $\mathbf{M}$ to the attention weights at every layer. $\mathbf{M}$ is of size $n \times n$, with $n$ being the size of the input sequence. Every element $\mathbf{M}_{ij}$, represents whether input $w_i$ should attend to $w_j$. Thus $M_{ij}$ equals 1 if both $w_i$ and $w_j$ belong to the same set of tokens (permissible/impermissible). Furthermore, as the hidden representation of [cls] is used for classification, no token can attend to [cls] while [cls] can attend to all tokens.

### 3.2.2   Sequence-to-sequence models

For the sequence-to-sequence tasks, Pruthi et al. use an encoder-decoder model employing different attention mechanisms. Both the encoder and the decoder consist of an embedding layer with embedding size 256, followed by a dropout layer with dropout ratio 0.5, a single layer bidirectional GRU introduced by Cho et al. (2014) with a hidden dimension of 512, and a linear layer with an input size of 2 * 512.

In addition to these components, the decoder contains a dot product attention mechanism which attends over all the hidden states produced by the encoder. Pruthi et al. also used two variants of this encoder-decoder model as baseline. One in which the attention mechanism only has uniform attention weights, and one with no attention. These baselines should clearify whether learned attention weights actually add a benefit for performing the task.

### 3.3   Datasets

The manipulability of three different binary classification tasks and three different sequence-to-sequence tasks were reproduced from Pruthi et al.. Furthermore an additional multiclass classification task is introduced to test if attention is still easy manipulable on harder tasks. All the datasets necessary to run the experiments, including the training, development, and test splits, are provided in their repository. GitHub repository[1].

### 3.3.1   Classification tasks

**Occupation classification**   De-Arteaga et al. (2019) introduced a dataset of biographies of different occupations. Pruthi et al. used the biographies of surgeons and physicians from the multiclass classification prediction setup to create a binary classification setup. Further on, the minority classes (female surgeons and male physicians) are down-sampled by a factor of ten, so that the number of surgeons and physicians in each gender are almost equal. The data with a ratio of one surgeon to two physicians is split into 17629 training samples, 2519 validation samples and 5037 test samples. In all these samples the set of impermissible tokens consists out of gender pronouns.

**Pronoun-based Gender Identification**   Pruthi et al. constructed a dataset from different Wikipedia biographies. Every biography had one type of gender distinct pronoun and that was used for automatically labelling the biographies on their gender. The dataset consists out of 11271 samples and is split into a test, validation and train set in a 8:1:1 ratio. Each split consists roughly out of as many man as woman. For training the base model without impermissible tokens, these gender distinct pronouns are replaced by neutral pronouns instead.

**Sentiment Analysis + Wikipedia sentences**   Pruthi et al. used the binary version of the Stanford Sentiment Treebank Socher et al. (2013). Pruthi et al. said to use 10523 movie reviews. After a recount of the dataset, the dataset contains 9613 movie reviews, where each review is classified as positive or negative. To each review, a random opening sentence of a Wikipedia page is appended. Combining these two sentences, the movie review is seen as the impermissible tokens. The data set is equally split into positive and negative examples, where the split ratio of training:validation:test is 8:1:2.

---

[1]`https://github.com/danishpruthi/deceptive-attention/tree/master/src/classification_tasks/data`

### 3.3.2 Sequence-to-sequence tasks

Below we describe three different synthetic datasets introduced by Pruthi et al. for sequence-to-sequence tasks. As Pruthi et al. created these datasets, the gold alignment for each instance in the dataset is known. These alignments are then used as impermissible tokens.

**Bigram Flipping**  The goal is to reverse the bigrams ($\{w_1, w_2 \ldots w_{2n-1}, w_{2n}\} \rightarrow \{w_2, w_1 \ldots w_{2n}, w_{2n-1}\}$)

**Sequence Copying**  The goal here is copy the input sequence ($\{w_1, w_2 \ldots w_{n-1}, w_n\} \rightarrow \{w_1, w_2 \ldots w_{n-1}, w_n\}$)

**Sequence Reversal**  The goal here is to reverse the input sequence ($\{w_1, w_2 \ldots w_{n-1}, w_n\} \rightarrow \{w_n, w_{n-1} \ldots w_2, w_1\}$)

For all three tasks 100,000 random input sequences, with a length up to 32 tokens, are generated for the training, validation and test set. The vocabulary of the output and input is set to 1000 unique tokens. For the bigram flipping, the input sequence is set to be even.

**Machine translation (English to German)**  For this task, the Multi30K dataset from Elliott et al. (2016) is used, which includes 29,000 training, 1014 validation and 1000 test samples. Because there exists no golden alignment between the two translations, the Fast Align toolkit Dyer et al. (2013) is used to create the alignment between the target words and their source counterparts. These alignments are then used as impermissible tokens.

### 3.3.3 Further extension

**Multiclass Sentiment Analysis + Wikipedia sentences**  The multiclass sentiment analysis is very similar to binary sentiment analysis but each review can now have one of five labels, ranging from 'very negative' to 'very positive'. The dataset consists out of the same 9613 reviews from the Stanford Sentiment Treebank. The data is split in the same train-, validation- and test-set as in the binary classification sentiment analysis.

### 3.4 Experimental setup and code

All the code ran in this report can be found in our GitHub repository[2]. It includes a Jupyter notebook, which gives a clear overview on reproducing all the results in this report. Furthermore, this repository includes code from the repository of Pruthi et al., slightly modified for use in the notebook.

### 3.4.1 Classification tasks

The code ran to obtain the results of this report was taken from the GitHub repository from Pruthi et al.. Each of the models, except for BERT, are trained and evaluated using five different random seeds (1 to 5). Due to a limited computational budget, we only trained and evaluated BERT using three different random seeds (42, 10, 20).

Each model was trained using three different values of $\lambda \in \{0, 0.1, 1\}$ and once where the impermissible data was anonymized/deleted and attention was not manipulated. The Embedding and LSTM models were trained for a maximum of 15 epochs, and BERT for five epochs. After each epoch, the models were evaluated on the evaluation set. If the performance of the model on the evaluation set did not improve for two epochs, training was halted. The validation accuracy was calculated and the training was stopped early if the validation accuracy did not increase after two epochs. When the model finished training, the test accuracy and the attention mass on impermissible tokens were saved for the model with the best validation accuracy. For the final result, the average of the test accuracy and attention mass over the different seeds was reported.

### 3.4.2 Sequence-to-sequence tasks

The three synthetic tasks and the machine translation task run in the encoder-decoder model on five different seeds (1 to 5), using three different values of $\lambda \in \{0, 0.1, 1\}$. To calculate the baseline of the tasks models with none and uniform attention were run, on the same seeds. They ran a max of 30 epochs. The model stops the training when the next epoch does not get a lower validation loss than the last epoch. For the three synthetic tasks, the test accuracy and the attention mass is saved. For the machine translation, the BLEU score[3] is calculated between the target sentences and the output sentences from the model.

---

[2]https://github.com/koengommers/fact-ai-2020
[3]https://github.com/neulab/compare-mt

### 3.5 Computational requirements

The BiLSTM and Embedded model were run on a NVIDIA GeForce GTX 1080 GPU. For the BiLSTM model, each epoch runs around 150 seconds for the occupation classification task and around 65 seconds for the other three tasks. The Embedded model runs each epoch for the occupation classification task for 85 seconds and for the other three tasks for 40 seconds.

BERT was trained on a NVIDIA GTX 1080TI GPU. Each training epoch for the Occupation classification dataset takes approximately 600 seconds to complete, whereas each epoch on the Gender Identification dataset takes approximately 300 seconds to complete.

The encoder-decoder model is run on the same GPU as the BiLSTM and Embedded model. All three synthetic tasks take around 720 seconds to complete one epoch with a batch size of 128, whereas the machine translation took around 180 seconds to complete one epoch.

## 4 Results

Table 1 shows the results of the classification tasks. We find that the proposed way of training models according to Pruthi et al. indeed decreases the attention mass on impermissible tokens. Although models assign little to no attention to these tokens, the models still achieve a higher accuracy on almost every task than models trained without impermissible tokens. Therefore, these results support the main claims made by Pruthi et al.. The same applies to the results for the sequence-to-sequence tasks shown in table 1, where the manipulated models have less attention and still maintain a higher accuracy than the none manipulated models.

### 4.1 Results reproducing original paper

#### 4.1.1 Classification Results

At first sight, our results from the classification experiments, presented in table 1, are in concordance with those from Pruthi et al.. For both the Occupation Prediction and Gender Identification tasks, it is visible that with values of $0.1$ and $1.0$ for $\lambda$, the assigned attention mass on impermissible tokens diminishes with minimal reduction in accuracy. Apart from the BERT baseline, depicted on the first row for both BERT variants, each of our reported accuracy scores for Occupation Prediction and Gender Identification differ no more than one percent than those reported by Pruthi et al..

The results for the baseline of each model, depicted on every first row, help support the second claim presented in section 1: Manipulated models still rely on information from impermissible tokens. By comparing the second row with the first row for each model, we see that when anonymising or deleting the impermissible tokens results in a drop in accuracy. This entails that the impermissible tokens contain information important for performing the different tasks. If a manipulated model, with of $0.1$ or $1.0$, results in a higher accuracy than the baseline, we can conclude that the model still uses information from the impermissible tokens.

For the BERT baseline, we report an accuracy of 82.9 percent, whereas Pruthi et al. report an accuracy of 72.8 percent. Despite this difference, we still see a considerable difference between our BERT baseline and its manipulated versions. Therefore, we can draw the same conclusion from these results, namely that the manipulated models still use information from the impermissible tokens.

For the SST + Wiki we report slightly different accuracy scores for the Embedding and BiLSTM models. For example, we report an accuracy of 68 percent for the BiLSTM with $\lambda = 0.1$, whereas Pruthi et al. report an accuracy of 60.6. However, there are two reasons why this does not necessarily invalidate the claims from section 1. First of all, the standard deviation of the accuracy scores for these models are higher for the SST + Wiki than for the Occupation Prediction and Gender Identification tasks. As Pruthi et al. do not report their obtained standard deviation between random seeds, we cannot dismiss that they also obtain larger standard deviations for this task, which would entail that our obtained results do not differ a lot from theirs. Secondly, Pruthi et al. speculate that the BiLSTM and Embedding models are under parameterized for the SST + Wiki task, as a result jointly reducing attention mass and retaining accuracy would be harder. This is something we also see in our results for these models, as the accuracy decreases when the attention mass gets reduced. Manipulating BERT on the other hand, does not result in a decrease in accuracy. Which is why we believe that the two claims presented in section 1 are still valid.

**Multiclass Sentiment Classification**   was added to investigate if the attention manipulation scheme described in section 3.1 generalises to more difficult tasks. As we can see in table 1, the Embedding and BiLSTM models do not

achieve much higher scores than the baseline which does not have access to relevant features for this task. Therefore, we can not regard the results of these models for this task as either affirmative nor invalidating.

For BERT, we can see that the results show a similar trend as for the other tasks. The baseline performs worse than the other versions with a difference of around $8$ percent. As this is higher than random guessing, which would be the baseline, we can conclude that the manipulated versions of BERT use information from the impermissible tokens. Furthermore, we see no decrease in accuracy with a reduced amount of attention mass on impermissible tokens. Therefore, these results show that both claims made in section 1 can be extended to a more difficult multiclass classification problem.

| Model | $\lambda$ | $\mathcal{I}$ | Occupation Pred. | | Gender Identify | | SST + Wiki | | SST + Wiki M.C. | |
| --- | --- | --- | --- | --- | --- | --- | --- | --- | --- | --- |
| | | | Acc ± Std | A.M. | Acc ± Std | A.M. | Acc ± Std | A.M. | Acc ± Std | A.M. |
| Embedding | 0.0 | ✗ | 93.4 ± 0.32 | - | 70.8 ± 1.55 | - | 48.6 ± 1.05 | - | 24.9 ± 0.49 | - |
| Embedding | 0.0 | ✓ | 96.5 ± 0.18 | 57.0 ± 2.42 | 100.0 ± 0.0 | 95.3 ± 1.27 | 71.7 ± 0.96 | 49.5 ± 2.11 | 26.3 ± 2.96 | 46.0 ± 2.80 |
| Embedding | 0.1 | ✓ | 96.3 ± 0.23 | 7.6 ± 2.81 | 100.0 ± 0.0 | 8.0 ± 0.60 | 70.1 ± 1.10 | 39.6 ± 1.11 | 26.3 ± 2.12 | 36.2 ± 0.88 |
| Embedding | 1.0 | ✓ | 96.2 ± 0.11 | 1.6 ± 0.31 | 99.8 ± 0.13 | 4.2 ± 1.33 | 50.5 ± 0.63 | 10.9 ± 3.68 | 24.6 ± 1.66 | 9.5 ± 1.29 |
| BiLSTM | 0.0 | ✗ | 93.6 ± 0.15 | - | 70.7 ± 0.53 | - | 52.9 ± 1.62 | - | 26.5 ± 0.88 | - |
| BiLSTM | 0.0 | ✓ | 96.6 ± 0.18 | 47.5 ± 5.81 | 100.0 ± 0.0 | 92.2 ± 5.07 | 76.9 ± 1.25 | 78.3 ± 4.86 | 28.9 ± 0.93 | 50.0 ± 2.0 |
| BiLSTM | 0.1 | ✓ | 96.6 ± 0.08 | 0.4 ± 0.24 | 99.9 ± 0.07 | 0.1 ± 0.14 | 68.0 ± 2.87 | 1.3 ± 1.08 | 26.8 ± 0.69 | 0.9 ± 0.43 |
| BiLSTM | 1.0 | ✓ | 96.6 ± 0.11 | 0.0 ± 0.0 | 99.9 ± 0.07 | 0.0 ± 0.0 | 60.5 ± 2.32 | 0.1 ± 0.09 | 26.9 ± 0.31 | 0.1 ± |
| BERT | 0.0 | ✗ | 95.8 ± 0.17 | - | 82.9 ± 0.25 | - | 50.5 ±0.58 | - | 28.4 ± 2.20 | - |
| BERT (mean) | 0.0 | ✓ | 97.1 ± 0.08 | 10.3 ± 9.60 | 99.5 ± 0.47 | 44.1 ± 3.67 | 91.6 ± 0.23 | 19.0 ± 4.35 | 36.5 ± 0.46 | 18.7 ± 6.76 |
| BERT (mean) | 0.1 | ✓ | 97.1 ± 0.16 | 0.0 ± 0.0 | 99.8 ± 0.05 | 0.0 ± 0.0 | 91.8 ± 0.36 | 0.1 ± 0.02 | 36.8 ± 1.17 | 0.2 ± 0.05 |
| BERT (mean) | 1.0 | ✓ | 97.3 ± 0.22 | 0.0 ± 0.0 | 99.9 ± 0.01 | 0.0 ± 0.0 | 91.5 ± 0.22 | 0.0 ± 0.0 | 36.1 ± 0.65 | 0.0 ±0.01 |
| BERT | 0.0 | ✗ | 95.8 ± 0.17 | - | 82.9 ± 0.25 | - | 50.5 ± 0.58 | - | 28.4 ± 2.20 | - |
| BERT (max) | 0.0 | ✓ | 97.1 ± 0.08 | 65.8 ± 55.50 | 99.8 ± 0.15 | 99.4 ± 0.11 | 91.4 ± 0.31 | 64.9 ± 4.86 | 36.5 ± 0.46 | 67.8 ± 6.48 |
| BERT (max) | 0.1 | ✓ | 97.3 ± 0.18 | 0.0 ± 0.0 | 99.9 ± 0.01 | 0.0 ± 0.0 | 91.2 ± 0.76 | 0.1 ± 0.03 | 36.6 ± 0.5 | 0.3 ± 0.04 |
| BERT (max) | 1.0 | ✓ | 97.3 ± 0.16 | 0.0 ± 0.0 | 99.9 ± 0.05 | 0.0 ± 0.0 | 91.5 ± 0.06 | 0.0 ± 0.0 | 37.3 ± 0.32 | 0.0 ± 0.02 |

Table 1: Accuracies and standard deviations of multiple classification models along with their attention mass (A.M.) on impermissible tokens $\mathcal{I}$, with varying values of the loss coefficient $\lambda$. The first row for each model class represents the case when impermissible tokens $\mathcal{I}$ for the task are deleted/anomynized. The results for the Embedding and BiLSTM model are averaged over 5 seeds, and the results for BERT are averaged over 3 seeds.

### 4.1.2 Sequence-to-sequence Results

In table 2, we can see that the results obtained from the classification tasks generalise to the sequence-to-sequence tasks. The amount of Attention Mass assigned to impermissible tokens successfully decreases for the manipulated models. All the while, the accuracy score does not decrease by more than $0.5$ percent. Since the amount of attention mass decreases, but the models still achieve roughly the same accuracy, thus the first claim in section 1 is supported by these results.

For these experiments, the model with Uniform and None attention mechanisms acts as a baseline to show that learned attention weights are beneficial for the performance on these tasks. The manipulated models outperform the models with none or uniform attention. Therefore we can conclude that the model has benefit from the learned attention weights. Furthermore, since for the Bigram Flip, Sequence Copy, and Sequence Reverse tasks, the gold alignments are known a-priori, we can also confirm that the manipulated model still uses information from the impermissible alignments for completing these tasks.

One important note is that we report high standard deviation in attention mass for the model with $\lambda = 0.1$ on the Sequence Copy and Sequence Reverse tasks. As Pruthi et al. do not report their obtained standard deviation between random seeds, we cannot directly compare these results with theirs. However, they do note that they found that different using different random seeds, resulted in different learned alignments. This could result in a high variance in attention mass assigned to the impermissible alignments.

| Model | $\lambda$ | Bigram Flip | | Sequence Copy | | Sequence Reverse | | En $\to$ De MT | |
| --- | --- | --- | --- | --- | --- | --- | --- | --- | --- |
| | | Acc. | A.M. | Acc. | A.M. | Acc. | A.M. | BLEU | A.M. |
| Dot-Product | 0.0 | 100.0 ± 0.0 | 93.7 ± 0.3 | 100.0 ± 0.1 | 94.0 ± 0.3 | 100.0 ± 0.0 | 94.0 ± 0.0 | 25.3 ± 1.1 | 24.3 ± 1.2 |
| Uniform | 0.0 | 92.5 ± 2.8 | 4.7 ± 0.0 | 81.8 ± 0.9 | 4.7 ± 0.0 | 83.6 ± 7.2 | 4.7 ± 0.0 | 17.9 ± 1.1 | 6.0 ± 0.0 |
| None | 0.0 | 94.9 ± 1.6 | 0.0 ± 0.0 | 83.8 ± 4.1 | 0.0 ± 0.0 | 90.8 ± 3.3 | 0.0 ± 0.0 | 15.4 ± 1.6 | 0.0 ± 0.0 |
| Manipulated | 0.1 | 100.0 ± 0.0 | 13.9 ± 5.5 | 100.0 ± 0.1 | 17.2 ± 17.5 | 100.0 ± 0.0 | 13.4 ± 11.4 | 24.2 ± 0.8 | 17.0 ± 3.1 |
| Manipulated | 1.0 | 99.5 ± 0.5 | 0.0 ± 0.0 | 99.8 ± 0.3 | 0.0 ± 0.0 | 99.9 ± 0.0 | 0.0 ± 0.0 | 21.5 ± 2.3 | 1.4 ± 0.8 |

Table 2: Performance of sequence-to-sequence models and their attention mass (A.M.) on impermissible tokens $\mathcal{I}$, with varying values of the loss coefficient $\lambda$. The results are averaged over 5 random seeds.

## 5 Discussion

In this report, we have successfully reproduced the paper of Pruthi et al.. Three binary classification experiments and four sequence-to-sequence experiments were run with the attention manipulation scheme presented by Pruthi et al.. With the results from these tasks we were able to conclude that attention weights can be manipulated without a large change in performance for a variety of tasks. We were also able to show that these manipulated models still relied on impermissible tokens while there was little to no attention mass assigned to these tokens. All in all, we have shown that two main claims made by Pruthi et al. can be supported by our results. Furthermore, with the added multiclass classification task, we have shown that the results also generalise to other and more difficult multiclass classification tasks.

The discussed claims raise issues for the transparency for models that are explained via attention. Pruthi et al. argue that attention cannot always be used as an explanation of model decisions, because attention weights can be manipulated with little cost in performance. Authors of papers or malicious practitioners can use this attention manipulation to mislead readers or stakeholders. Therefore authors trying to explain decisions made by the model should not solely rely on attention weights of the model, but they should provide additional form(s) of explanation to provide insights in the inner workings of their models.

Overall the paper was easy to reproduce, because the code for large parts of the experiments was published in a GitHub repository. We contacted Pruthi et al. about missing code for running the experiments with BERT. We got a fast response, and within a couple days, code for running these experiments was added to the repository. Only the code for reproducing the baseline for BERT was missing, therefore we had to slightly extend their code to reproduce these results. Also, instructions on how to run the experiments for the baseline of the BiLSTM and Embedding model was not provided. We have added these instruction to our code repository.

For future work, more tasks can be explored to see if these claims can be generalised on more tasks and different domains. Attention mechanisms are also used for visual tasks such as image captioning Xu et al. (2015), and more recently the use of transformers on images Dosovitskiy et al. (2020).

Another question which should be further researched arises from this paper, "If attention is not always a good explanation of the decisions made by a model, what is?". Kobayashi et al. (2020), argue that the vector norms of transformed input vectors of attention mechanisms play a role in the output of the model. It could be investigated whether an additive penalty on the vector norms of impermissible tokens will affect the performance of attention based models.

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
