# OpenReview forum: "[Rp] Learning to Deceive with Attention-Based Explanations"
_ML_Reproducibility_Challenge/2020 — Reject_

### Official Review · AnonReviewer2 · 2021-03-01
**Good effort in reproducing the work**

**Rating:** 7
**Confidence:** 3

**Review:**

The report described the authors' efforts in reproducing the work "Learning to Deceive with Attention-Based Explanations". The report is well written in general. The authors have reproduced most of the experiments from the original paper, except for the ones requiring a private dataset, which is understandable. In place of the missing experiment, the authors added a new result with a multi-class classification problem, which supplements the original results. In addition, the authors reported and analyzed the variance of the proposed methods, which were not present in the original paper. The discussion of the report is also insightful.

Though I think the report did a good job in overal, one aspect where it can be further improved is that the hyper-parameters used in the experiments were not discussed: did the authors used the same ones from the original work? is the proposed method sensitive to the selection of different hyper-parameters? Some investigations in this  direction would be valuable.

**Familiar With The Original Paper:**

I have not read the original paper

**Reproducibility Summary:**

Report has summary

---

### Official Review · AnonReviewer3 · 2021-03-01
**Excellent work**

**Rating:** 9
**Confidence:** 4

**Review:**

The authors performed well. They had good communication with the original authors and was able to substitute for missing dataset.

**Familiar With The Original Paper:**

I have not read the original paper

**Reproducibility Summary:**

Report has summary

---

### Decision · Program_Chairs · 2021-03-31

**Decision:**

Reject

**Comment:**

Overall reviews and/or the paper content not good enough for the AC to recommend to the journal.